# Effectiveness and Safety of EGFR-TKI Rechallenge Treatment in Elderly Patients with Advanced Non-Small-Cell Lung Cancer Harboring Drug-Sensitive EGFR Mutations

**DOI:** 10.3390/medicina57090929

**Published:** 2021-09-03

**Authors:** Yutaka Yamada, Hisao Imai, Tomohide Sugiyama, Hiroyuki Minemura, Kenya Kanazawa, Takashi Kasai, Koichi Minato, Kyoichi Kaira, Takayuki Kaburagi

**Affiliations:** 1Department of Respiratory Medicine, Ibaraki Prefectural Central Hospital, Kasama 309-1793, Ibaraki, Japan; t-kaburagi@chubyoin.pref.ibaraki.jp; 2Department of Respiratory Medicine, Comprehensive Cancer Center, International Medical Center, Saitama University Hospital, Hidaka 350-1298, Saitama, Japan; m06701014@gunma-u.ac.jp (H.I.); kkaira1970@yahoo.co.jp (K.K.); 3Division of Respiratory Medicine, Gunma Prefectural Cancer Center, Ota 373-8550, Gunma, Japan; kminato@gunma-cc.jp; 4Division of Thoracic Oncology, Tochigi Cancer Center, Utsunomiya 320-0834, Tochigi, Japan; tomsugiy@tochigi-cc.jp (T.S.); takasai@tochigi-cc.jp (T.K.); 5Department of Pulmonary Medicine, Fukushima Medical University, Fukushima 960-1295, Japan; hiromine1129@gmail.com (H.M.); k-kenya@fmu.ac.jp (K.K.); 6Clinical Oncology Center, Fukushima Medical University Hospital, Fukushima 960-1295, Japan

**Keywords:** epidermal growth factor receptor, advanced non-small-cell lung cancer, elderly patients, re-administration, tyrosine kinase inhibitor, EGFR-TKI, *EGFR* mutations, secondary chemotherapy

## Abstract

*Background and Objectives*: Epidermal growth factor receptor–tyrosine kinase inhibitors (EGFR-TKIs) are effective first-line chemotherapeutic agents for patients with advanced non-small-cell lung cancer (NSCLC) harboring drug-sensitive *EGFR* mutations. However, the effectiveness of EGFR-TKI rechallenge after first-line EGFR-TKI treatment is not sufficient in elderly patients (over 75 years of age) harboring drug-sensitive *EGFR* mutations. Therefore, we investigated the effectiveness and safety of EGFR-TKI rechallenge after first-line EGFR-TKI treatment in elderly patients with advanced NSCLC harboring drug-sensitive *EGFR* mutations. *Materials and Methods*: Between April 2008 and December 2015, we analyzed 78 elderly patients with advanced NSCLC harboring drug-sensitive *EGFR* mutations with first-line EGFR-TKI treatment at four Japanese institutions. We retrospectively evaluated the clinical effectiveness and safety profiles of EGFR-TKI rechallenge after first-line EGFR-TKI treatment in elderly patients with advanced NSCLC harboring drug-sensitive *EGFR* mutations (exon 19 deletion/exon 21 L858R mutation). *Results*: Twenty-two patients in the cohort were rechallenged with EGFR-TKI. The median age was 79.5 years (range 75–87 years). Despite the fact that it was a retrospective analysis, even with EGFR-TKI rechallenge treatment the response rate was 23%, progression-free survival was 5.3 months, and overall survival was 14.4 months. Common adverse events included rash acneiform, paronychia, diarrhea, and anorexia. There were no treatment-related deaths. Due to the occurrence of adverse events of grade 2 or more, dose reduction was performed in 15 (68.2%) of 22 cases. *Conclusions*: EGFR-TKI rechallenge treatment after first-line EGFR-TKI treatment in elderly patients with advanced NSCLC harboring drug-sensitive *EGFR* mutations was one of the limited, safe and effective treatment options for elderly *EGFR*-positive lung cancer patients.

## 1. Introduction

Population aging and progress of cancer treatment increased the number of elderly lung cancer patients [1]. More than 50% of lung cancer patients are diagnosed at over 65 years old. This is the lower limit to define the ‘elderly’ in epidemiological studies [2]. In addition, 85% of adult lung cancer consists of non-small cell lung cancer (NSCLC) [3]. As first-line treatment for patients with advanced NSCLC harboring drug-sensitive *EGFR* mutations, previous clinical trials showed the effectiveness of epidermal growth factor receptor–tyrosine kinase inhibitors (EGFR-TKIs) such as gefitinib, erlotinib and afatinib [4,5,6,7,8,9]. Furthermore, in elderly patients with advanced NSCLC harboring *EGFR* mutations, gefitinib and erlotinib as a first-line treatment provided high response rate (RR) and long-term survival [10,11,12,13,14,15].

In *EGFR* gene mutation-positive patients, there are no guidelines for determining the order of administration of EGFR-TKI monotherapy and cytotoxic anticancer agents. However, for patients with *EGFR* mutation-positive cancer, it is recommended not to miss treatment by EGFR-TKI single agent. The main toxicities associated with EGFR-TKI are diarrhea, skin rash, paronychia, and hepatic dysfunction. Although the frequency and severity of toxicity are different for each drug, it has been suggested that drug suspension and dose reduction enable long-term treatment continuation [6,7,10,16,17,18].

For patients treated with single agent gefitinib or erlotinib as the first-line treatment, if a biopsy is positive for the *EGFR* T790M mutation, osimertinib can be given as a single agent for the second-line treatment. Nevertheless, T790M negative cases after re-biopsy are currently being treated according to the first-line treatment of NSCLC without driver mutation (e.g., EGFR, ALK (anaplastic lymphoma kinase), ROS1).

In clinical practice, administration of osimertinib was previously required to obtain T790M-positive results by biopsy. However, in the elderly, it may be difficult to re-biopsy. Therefore, it is common practice for *EGFR*-positive elderly patients to receive cytotoxic chemotherapy as second-line treatment, or to re-administer a different EGFR-TKI as a subsequent treatment.

The significance of EGFR-TKI re-administration in patients who relapsed after first-line EGFR-TKI treatment has been previously reported [18], but there have been no reports specifically on the elderly. To explore this clinical question, we conducted a retrospective study of the effectiveness, safety, and proportion of EGFR-TKI rechallenge treatment after first-line EGFR-TKI received in elderly patients (over 75 years of age) with advanced NSCLC harboring *EGFR* mutations.

This study was conducted additionally to a previous study by Imai et al., “Efficacy and safety of cytotoxic drug chemotherapy after first-line EGFR-TKI treatment in elderly patients with non-small-cell lung cancer harboring sensitive EGFR mutations” [19]. 

## 2. Materials and Methods

### 2.1. Patients

This study included 78 elderly patients with advanced NSCLC harboring drug-sensitive *EGFR* mutations who were treated with EGFR-TKI as first-line treatment at four Japanese institutions between April 2008 and December 2015. The histological diagnosis and staging of NSCLC were based on the classification of the World Health Organization and the system of TNM staging, respectively [20]. Eligibility criteria were histologically or cytologically confirmed NSCLC, unresectable stage III/IV disease, and *EGFR* mutation (exon 18 G719X, exon 19 deletion or exon 21 L858R mutation). All patients were untreated with EGFR-TKIs and received first-line gefitinib (250 mg/day) or erlotinib (150 mg/day) and were then subsequently treated with other EGFR-TKIs (gefitinib, erlotinib, afatinib, or osimertinib). Treatments after the second-line treatment were determined by the treating physician. This continued until disease progression, intolerable toxicity, or withdrawal of treatment consent. To clarify EGFR-TKI rechallenge treatment performed in elderly patients with advanced *EGFR* mutation-positive advanced NSCLC, we collected cases in which the first-line treatment was gefitinib or erlotinib monotherapy, and finally EGFR-TKI rechallenge was performed.

A clinical chart search for eligible cases was conducted at each hospital. Records for baseline characteristics, chemotherapy regimens and responses to every EGFR-TKI treatment were also collected. This study was approved by an institutional review board of each institution. 

### 2.2. EGFR Mutation Analysis

*EGFR* mutations were analyzed with allele specific real-time polymerase chain reaction (PCR) using biopsy or cytology specimens in each institution. The method for extracting genomic DNA is as previously described [21,22].

### 2.3. Response Evaluation

The best overall response and maximum tumor contraction were recorded as tumor responses. Tumor responses were classified as complete (CR), partial (PR), stable disease (SD), progressive disease (PD), or not evaluable (NE), according to the response evaluation criteria in solid tumors (RECIST, version 1.1) [23]. 

### 2.4. Statistical Analysis

Fisher’s exact test, the Chi-squared test and the Mann-Whitney U test were used to compare patient characteristics. The Kaplan–Meier method was used to estimate progression-free survival (PFS) and overall survival (OS). Adverse events of EGFR-TKI rechallenge were evaluated by the Common Terminology Criteria for Adverse Events (CTCAE), version 4.0. All statistical analyses were performed using SAS, version 9.4 (SAS Institute, Cary, NC, USA).

## 3. Results

### 3.1. Patient Selection and Characteristics

Figure 1 summarizes the number of patient selections and evaluation data at each time point. 78 qualified elderly patients with advanced NSCLC received first-line treatment with EGFR-TKIs. After failure of first-line EGFR-TKI treatment, patients were allowed subsequent treatment due to their own consensus, including continuation of EGFR-TKI treatment. Among these, 44 (54.3%) received second-line chemotherapy, and 14 out of these 44 patients received rechallenge with EGFR-TKIs. Eight of the 14 patients who received second-line chemotherapy subsequently received EGFR-TKIs as third-line treatment. Consequently, a total of 22 patients were included in the EGFR-TKI rechallenge analysis.

Table 1 shows the characteristics of patients treated with EGFR-TKI rechallenge treatment. After starting first-line EGFR-TKI treatment, median PFS was 5.3 months and median OS was 14.4 months. The median follow-up was 14.6 months (range 3.2–31.2 months; Figure 2). Most of the subjects surveyed (82%) had received administration of erlotinib after initial gefitinib treatment. In one case, gefitinib was re-administered after initial gefitinib treatment, and in another case erlotinib was re-administered after initial erlotinib treatment. However, in these two cases, a cytotoxic drug chemotherapy regimen was administered during EGFR-TKI rechallenge, not beyond PD.

### 3.2. Treatment Efficacy

Table 2 shows the objective tumor response to the first-line EGFR-TKI and the rechallenge after the first-line EGFR-TKI. None of the patients showed CR on EGFR-TKI rechallenge treatment, but 5 had PR, 12 had SD, and 4 had PD. Therefore, the overall RR was 23.0% and disease control rate was 77.0%.

Table 3 shows the types of second-line chemotherapy regimen. The RR and disease control rates (DCC) by *EGFR* mutations were 20% and 90% for exon 19 del, respectively, and 18.1% and 63.6% for exon 21 L858R, respectively. Furthermore, when the cohort was divided into those over or under 80 years of age, the RR and DCC rates were 27.3% and 90.9%, respectively, for those under 80 years of age, and 18.2% and 63.6%, respectively, for those over 80 years of age. Ten patients received EGFR-TKI re-rechallenge regimens. Seven of these cases were first- or second-generation EGFR-TKI re-rechallenge cases. Only two patients were treated with immune checkpoint inhibitor after EGFR-TKI rechallenge. In the case of PD after EGFR-TKI rechallenge of first- and second-generation EGFR-TKI, re-biopsy was performed in four of 22 cases, and three of these were T790M-positive. Osimertinib was subsequently administered as treatment in these three cases.

### 3.3. Survival Analysis

PFS after rechallenge with EGFR-TKI was shorter than after first-line treatment of EGFR-TKI (log-rank, *p* < 0.05). PFS during first-, second- and third-line treatment, and PFS after second-line, in the overall population are shown in Figure 3.

### 3.4. Toxicity and Adverse Events

The toxicity of the EGFR-TKI rechallenge was evaluated in all patients, and the main adverse events during EGFR-TKI rechallenge are summarized in Table 4. Common adverse events were rash acneiform (total 14/22, 63.63%; Grade 1, seven cases (31.81%); Grade 2, seven cases (31.81%)), paronychia (total 8/22, 36.36%; Grade 2, eight cases); diarrhea (total 9/22, 40.9%; Grade 1, four cases (18.18%); Grade 2, two cases (9.09%); Grade 3, one case (4.54%)); and anorexia (total 5/22, 22.72%; Grade 1, four cases )18.18%); Grade 2, one case (4.54%)). There were no CTCAE Ver4.0 Grade 4 or higher adverse events in the 22 cases. Grade 3 adverse events were diarrhea (*n* = 1) and infection (*n* = 1). The total number of adverse events was 22 (Grade 1, 12 cases (54.5%); Grade 2, eight cases (36.4%): Grade 3, two cases (9.09%)). Furthermore, Grade 1 Anemia was observed in only three cases (1.4%). The increase in AST/ALT was four cases (18.2%) in Grade 1, and one case (4.5%) in Grade 2, respectively.

### 3.5. Dose Change of EGFR-TKI

Due to the occurrence of adverse events of grade 2 or more, dose reduction was performed at 15 (68.2%) of 22 cases. In 8 cases (53.3%) of these 15 cases, there were further dose changes such as dose reduction or dose increase. In total, six cases (40.0%) received second-line dose reduction. Furthermore, of the above 15 cases, four (26.3%) responded to the increase in dose.

### 3.6. Relationship with Body Surface Area

The median body surface area (BSA) was 1.38 m^2^, and the average value was 1.44 m^2^. All cohorts showed very weak negative correlation (R = −0.35). When the cohort was divided into those with BSA greater or less than 1.4 m^2^, there was no correlation below 1.4 m^2^ (R = 0.02) and there was a very weak negative correlation above 1.4 m^2^ (R = −0.34) (Figure 4). The median PFS was 9.8 months in the low BSA group. In contrast, the median PFS was 3.5 months in the large BSA group, and median OS was 14.7 months in the low BSA group and 12.7 months in the large BSA group, respectively (Figure 5).

### 3.7. Figures, Tables and Schemes

EGFR-TKI, epidermal growth factor receptor–tyrosine kinase inhibitor.

Most Performance status at the start of treatment was 0–1. Most of the subjects surveyed (82%) had administration of erlotinib after initial gefitinib treatment.

None of the patients showed CR on EGFR-TKI rechallenge treatment. Therefore, the overall RR was 23.0% and disease control rate was 77.0%.

The BSC was the most selected after failure of EGFR-TKI rechallenge, and there were many cases of pemetrexed and re-rechallenge.

The table above lists all adverse events. There were no Grade 4 or higher adverse events in the 22 cases.

Common adverse events included rash acneiform, paronychia, diarrhea, and anorexia.

When the cohort was divided into those with BSA greater or less than 1.4 m^2^, there was no correlation below 1.4 m2 (R = 0.02) and there was a very weak negative correlation above 1.4 m^2^ (R = −0.34).

The median PFS was 9.8 months in the low BSA group. In contrast, the median PFS was 3.5 months in the large BSA group, and median OS was 14.7 months in the low BSA group and 12.7 months in the large BSA group, respectively.

## 4. Discussion

In this study, about 28.2% of the elderly patients with NSCLC harboring drug-sensitive *EGFR* mutations underwent EGFR-TKI rechallenge after first-line EGFR-TKI treatment. Our study suggests that EGFR-TKI rechallenge treatment in elderly patients is safe and effective, and adverse events were tolerable.

Research such as LUX-LUNG1 has indicated that EGFR-TKI rechallenge does not contribute to OS [24]. On the other hand, there are reports on the utility of EGFR-TKI rechallenge, particularly rechallenge after previous gefitinib treatment. In cases where the therapeutic effect of EGFR-TKI treatment was helpful, rechallenge can be an option when there is no other treatment choice [25,26,27,28,29].

Currently, standard second-line and subsequent treatments have not been established, especially for elderly patients with advanced NSCLC. Therefore, the effectiveness of re-challenge treatment in elderly patients with advanced NSCLC harboring *EGFR* mutations remains unclear. There are several treatment choices for elderly patients with advanced NSCLC. They are BSCs and third-generation drug monotherapy or non-platinum based or platinum-based combination chemotherapy, EGFR-TKI. However, the selection of treatment for elderly patients is limited by the performance status.

Previously, Inoue et al. reported that 37% of non-elderly patients with EGFR-mutated NSCLC received second- or subsequent EGFR-TKI rechallenge after first-line gefitinib [30]. In this study, because the subjects were elderly, only 22 (28%) of the 78 patients received EGFR-TKI rechallenge treatment after failure of first-line EGFR-TKI treatment. However, there are limited studies on elder NSCLC patients harboring *EGFR* mutations after subsequent treatment. In a study by Kuwako et al. [15], only 8 (13%) of the 62 patients who relapsed subsequently received EGFR-TKI rechallenge. Moreover, 11 patients received second-line chemotherapy (18%), which was more than those who received EGFR-TKI rechallenge treatment. Most (42/62; 68%) chose treatment with the best supportive care available. In our cohort, more than 50% of patients received second-line treatment and as many as 10 (13%) received third-line treatment. Among these, the number of cases of EGFR-TKI re-challenge was 22 (28%), which was higher than that of Kuwako’s study. Low-dose treatment of erlotinib has been reported to be both safe and effective, especially in elderly and frail patients [31].

In elderly patients, due to the presence of age-related organ dysfunctions and potentially complex complications, there is low resistance to chemotherapy toxicity as compared to younger patients. Therefore, the choice of cytotoxic chemotherapy is often withheld.

In clinical practice, the proportion of second-line cytotoxic chemotherapy for elderly patients with *EGFR* mutated NSCLC might be less likely than for adult patients.

Monotherapy of cytotoxic drug is currently standard first-line treatment for elderly patients with NSCLC. Kudo et al. reported cytotoxic drug monotherapy for elderly patients with advanced NSCLC in Japan. In this study, vinorelbine or docetaxel are selected for monotherapy, RR is 10 to 23%, PFS is 3.1 to 5.5 months, and OS is 9.9 to 14.0 months, respectively [32]. Although the number of patients in our study was smaller than in previous studies, our results were equivalent to the outcome of first-line treatment for elderly patients with metastatic NSCLC.

Most patients with NSCLC eventually become refractory to treatment, despite a good response to first-line treatment for EGFR-TKIs. Most treatment resistance is due to acquired *EGFR* mutations (i.e., T790M) or amplification of the MET oncogene [33].

Osimertinib, a third-generation EGFR-TKI, shows a better safety profile with monotherapy and is particularly useful for EGFR-TKI-resistant NSCLC with T790M mutations [34]. However, it is necessary to detect T790M by re-biopsy after using EGFR-TKI prior to administration of Osimertinib. For this reason, re-biopsy is important for deciding the next treatment. Unfortunately, in this study, most patients died before the T790M was evaluated, and the status of the T790M in most patients was unclear. Currently, T790M can be detected by repeating liquid biopsy instead of tissue re-biopsy. The expression rate of T790M due to repetition of tissue re-biopsy and liquid biopsy is different depending on the report [35,36,37,38,39,40]. However, the expression frequency of T790M in the re-biopsy of elderly patients is unclear.

Moreover, EGFR-TKIs should be able to switch to another treatment, or decide whether it should continue at the time of treatment failures. Continuation of gefitinib after radiological disease progression on first-line gefitinib did not prolong PFS in patients who received platinum-based doublet chemotherapy as subsequent line of treatment [41].

As mentioned above, elderly patients have many comorbidities and organ dysfunction as compared to young patients.

Thus, the toxicity associated with the treatment of elderly patients is an important issue. However, adverse events of EGFR-TKI rechallenge of elderly patients in this study were mild and predictable and their incidence and severity adverse events were similar to those observed in patients who received first-line treatment [32,42,43]. Moreover, few grade 3 events were observed and no treatment-related deaths were observed. Regardless of the severity, adverse hematological and non-hematological events were controllable.

Consequently, 82% of patients in this study were administered erlotinib after initial gefitinib treatment. The second-generation EGFR-TKI, erlotinib, was used in all cases in this study. In contrast, only three out of 22 patients received afatinib.

This is probably because the study period was from 2008 to 2015, and afatinib only became available in Japan in 2014.

In a Japanese study, Tamiya et al. reported that PFS was prolonged compared to the osimertinib-administered group after afatinib administration, as well as the osimertinib-administered group after gefitinib/erlotinib administration [44].

However, T790M mutation must be positive to allow osimertinib to be used after afatinib, and it has not been determined as the treatment method of T790M-negative cases after re-biopsy.

There is some prior research on EGFR-TKI and BSA or dose change, many of which are Japanese reports, and are limited to the studies of gefitinib or afatinib [45,46]. The cutoff value of the BSA in the previous report was 1.45–1.57 m^2^. In this study, 1.4 m^2^ became the cut-off value because the subject patients tended to be physically small. Furthermore, PFS or OS was prolonged when BSA was small as in the prior study, but there is no report that suggests the relationship with BSA for rechallenge, as in this study. Moreover, in this study, most of the agents that had a dose change for erlotinib, and the dose reduction in the elderly of smaller physique is considered to contribute to toxic control and treatment continuity. In other words, dose reduction may contribute to the continuation of treatment in elderly Japanese with deterioration of organ function and a smaller BSA.

In recent years, treatment methods for each *EGFR* mutation have been proposed, and especially for the exon21 L858R mutation, and due to the therapeutic effects of erlotinib and ramucirumab combination treatment, erlotinib has been attracting attention again [47]. Ramucirumab, a vascular endothelial growth factor (VGEF) inhibitor, has been used in combination with the treatment in this study. However, because it is generally difficult to administer to the elderly, erlotinib and ramucirumab combination treatment is not a treatment option for them.

Although it is in the clinical trial stage, third-generation EGFR-TKIs, nazartinib (EGF816) [48], lazertinib (YH25448) [49], abivertinib (AC0010) [50], and almonertinib (HS 10296) [51] are currently undergoing trials as first-line treatments. However, up to the present, the results for osimertinib and the extent of effectiveness have not been obtained.

There are several limitations to this study. First, this is a retrospective study for the selected patients. Second, EGFR-TKI selection is determined by the attending physician, and there might be selection bias in these decisions. Moreover, as a result, the choice can affect the survival after second-line treatment. The final limitation is a comparatively small population, which could not show statistical significance. To verify our findings for validity in clinical practice, there is a need for a larger prospective study.

## 5. Conclusions

In conclusion, our results indicate that EGFR-TKI rechallenge after first-line EGFR-TKI treatment in elderly patients with advanced NSCLC harboring drug-sensitive *EGFR* mutations was effective and tolerable. Approximately 28% of the elderly patients received EGFR-TKI rechallenge after first-line EGFR-TKI treatment. Despite the retrospective design, our findings show that EGFR-TKI rechallenge should be considered as a standard treatment after failure of EGFR-TKI in elderly patients with advanced NSCLC with T790M-negative *EGFR* mutations. Lastly, for elderly Japanese with a smaller BSA, dose reduction of EGFR-TKI might be a useful factor for continuation of treatment.

## Figures and Tables

**Figure 1 medicina-57-00929-f001:**
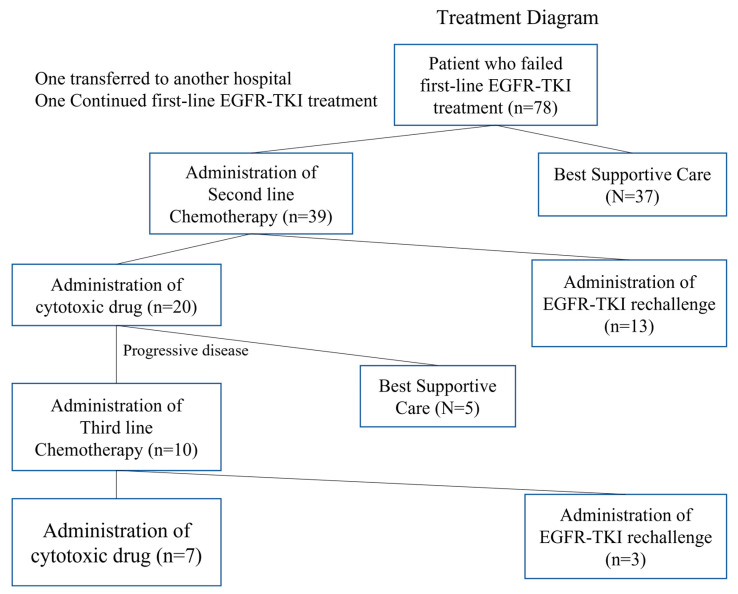
Study flow chart and patient treatment.

**Figure 2 medicina-57-00929-f002:**
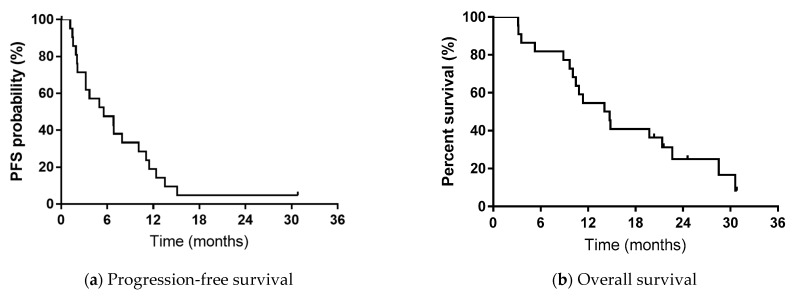
Progression-free survival and overall survival in patients treated with EGFR-TKI rechallenge. (**a**) Kaplan-Meier plots showing progression-free survival (PFS). Median PFS 5.5 months. (**b**) Kaplan-Meier plots showing overall survival (OS). The median OS is 14.4 months.

**Figure 3 medicina-57-00929-f003:**
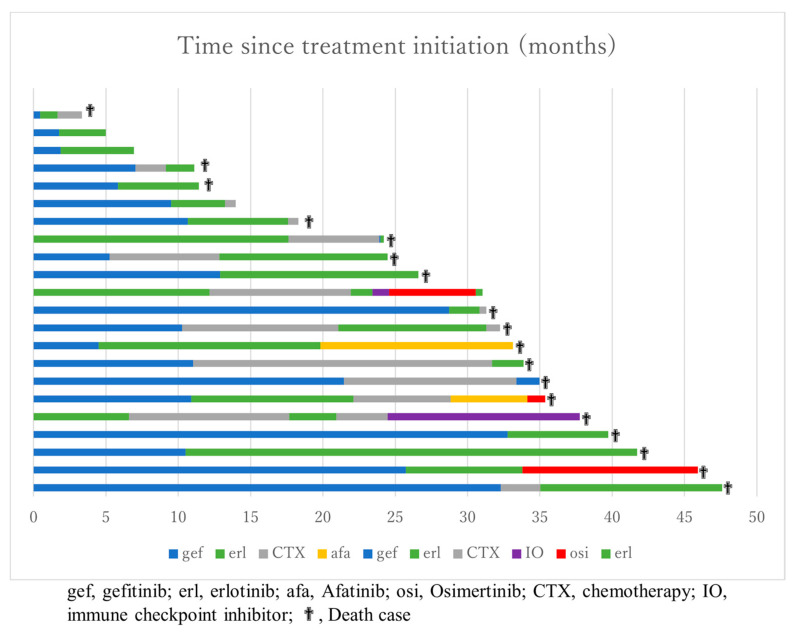
Swimmer plots of 22 patients who received EGFR-TKI rechallenge treatment. Individual swimmer plots display duration of treatment methods.

**Figure 4 medicina-57-00929-f004:**
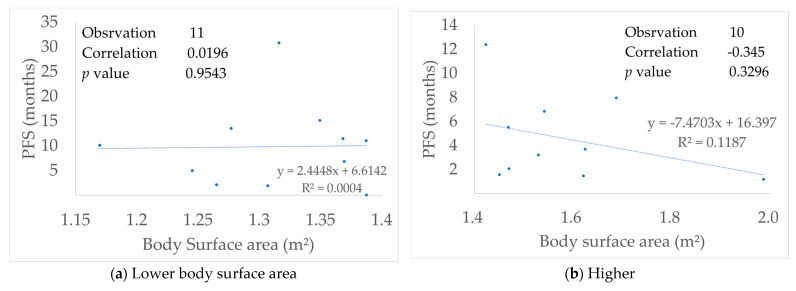
Correlation between body surface area and progression-free survival (PFS).

**Figure 5 medicina-57-00929-f005:**
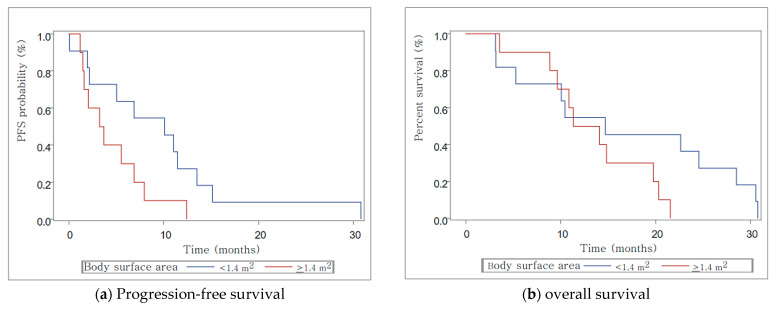
Progression-free survival (PFS) and overall survival (OS) of body surface area.

**Table 1 medicina-57-00929-t001:** Patients characteristics at initiation of EGFR-TKI rechallenge.

Characteristics	Number of Patients	(%)
Sex		
Male	7	32
Female	15	68
Age (years), median (range)	79.5 (75–87)	
Performance status		
0	7	32
1	11	50
2	4	18
3	0	0
4	0	0
Clinical Stage at diagnosis		
IIIB	1	4
IV	20	92
Postoperative recurrence	1	4
Histology		
Adenocarcinoma	22	100
Other/not otherwise specified	0	0
Smoking history		
Current or former	6	28
Never	15	68
Unknown	1	4
*EGFR* mutation		
Exon 19 deletion	10	45
Exon 21 L858R	11	50
Exon 18 G719X	1	4
First-line EGFR-TKI		
Gefitinib	19	86
Erlotinib	3	14
Afatinib	0	0
Rechallenge EGFR-TKI		
Gefitinib→Gefitinib	1	4
Gefitinib→Erlotinib	18	82
Erlotinib→Erlotinib	1	4
Erlotinib→Gefitinib	2	9
Administration line of rechallenge EGFR-TKI		
2	13	59
3	5	22
4	4	18
T790M mutation		
Positive	3	14
Negative	1	4
Unknown	18	72
Median follow-up period [months] (range)	14.6 (3.2–31.2)	

**Table 2 medicina-57-00929-t002:** Response to first-line EGFR-TKI and rechallenge EGFR-TKI in patients with EGFR-mutated non-small-cell lung cancer and treatment delivery.

	Number of Patients (%)	
	Primary EGFR-TKI	Rechallenge EGFR-TKI
Complete response	0 (0)	0 (0)
Partial response	17 (77)	5 (23)
Stable disease	1 (4)	12 (54)
Progressive disease	4 (19)	4 (19)
Not evaluable	0 (0)	1 (4)
Response rate (%)	77	23
Disease control rate ^a^ (%)	81	77
Dose reduction or alternative day administration		
Yes/No	10/12	15/7

^a^ The disease control rate is calculated as the number of patients with complete, partial, and stable disease divided by the total study population.

**Table 3 medicina-57-00929-t003:** Types of chemotherapy regimen between primary EGFR-TKI treatment and subsequent treatment.

Regimen	Number of Patients	
	Therapies before EGFR-TKI Rechallenge	Therapies after Failure of EGFR-TKI Rechallenge
DTX	5	0
PEM	3	7
S-1	3	1
CDDP-based combination chemotherapy	0	0
CBDCA-based combination chemotherapy	2	0
Immune check point inhibitor	0	2
EGFR-TKI rechallenge (Gefitinib, erlotinib, or afatinib)	-	7
Osimertinib *	-	3
Best supportive care	-	9

EGFR-TKI, epiderDTX, docetaxel; PEM, pemetrexed; CDDP, cisplatin; CBDCA, carboplatin. * Osimertinib was administered to three T790M-positive patients on re-biopsy after rechallenge of first- and second-generation EGFR-TKIs.

**Table 4 medicina-57-00929-t004:** Adverse events associated with EGFR-TKI rechallenge in elderly patients with epidermal growth factor receptor-mutated non-small cell lung cancer.

	NCI-CTCAE Grade (Ver 4.0)					
	1	2	3	4	5	≥3 (%)
Nonhematologic adverse events						
Fatigue	2	0	0	−	−	0 (0)
Paronychia	0	8	0	−	−	0 (0)
Pruritus	2	0	0	−	−	0 (0)
Rash acneiform	7	7	0	0	0	0 (0)
Dyspnea	3	0	0	0	0	0 (0)
Anorexia	4	1	0	0	0	0 (0)
Diarrhea	4	2	1	0	0	1 (4.5)
Mucositis	1	3	0	0	0	0 (0)
Nausea	2	0	0	0	0	0 (0)
Pain	1	0	0	−	−	0 (0)
Infection	0	0	1	0	0	1 (4.5)
Pneumonitis	0	0	0	0	0	0 (0)
Thromboembolic event	0	1	0	0	0	0 (0)
Hematologic or laboratory adverse events						
Anemia	3	0	0	0	0	0 (0)
ALT increased	4	1	0	0	−	0 (0)
AST increased	4	1	0	0	−	0 (0)

AST, aspartate aminotransferase; ALT, alanine aminotransferase.

## Data Availability

The data are not publicly available due to localization in the hospital.

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
