# Peer review of "Effectiveness and Safety of EGFR-TKI Rechallenge Treatment in Elderly Patients with Advanced Non-Small-Cell Lung Cancer Harboring Drug-Sensitive EGFR Mutations"

_medicina, 2021, doi:10.3390/medicina57090929_

Round 1
Reviewer 1 Report
Yamada and colleagues investigated the effectiveness and safety of EGFR-TKI rechallenge after first-line EGFR-TKI treatment in 78 elderly patients with advanced NSCLC with drug-sensitive EGFR mutations at four different sites within Japan. In this retrospective study, 22 patients were rechallenged with EGFR-TKI and the response rate was 23%, progression-free survival was 5.3 months, and overall survival was 14.4 months. Overall, this is an interesting topic for the readers. However, there are several presentational issues that need addressing. These items are listed below:
- Figure 1 needs to be remade as it is poorly presented. Background boxes make it difficult to follow
- It is almost impossible to read any of the detail in figure 2b. Please show Figure 2B with the same font size as figure 2a. This is also true for figures 4 and 5. Please make the figures larger and increase font size. Figure 4 would benefit from the inclusion of a correlation curve
- Table 3 overlaps with the line numbers and is difficult to see the details in the table
- It would be beneficial to perform descriptive statistics for most of the data in the tables
- There are several grammar issues throughout the manuscript
Author Response
Dear Reviewer 1
Thank you very much for reviewing our manuscript.
We wish to express our appreciation to the Reviewer for his or her insightful comments, which have helped us significantly improve this manuscript.
We have addressed your comments with point-by-point responses and revised the manuscript accordingly.
Responses to the Comments by the reviewer 1:
Point 1: Figure 1 needs to be remade as it is poorly presented. Background boxes make it difficult to follow
Response 1: As you pointed out, it is difficult to understand Figure 1, so we were significantly revised.
Point 2: It is almost impossible to read any of the detail in figure 2b. Please show Figure 2B with the same font size as figure 2a. This is also true for figures 4 and 5. Please make the figures larger and increase font size. Figure 4 would benefit from the inclusion of a correlation curve
Response 2: These figures have changed the font size. Also, we have added the correlation curve with respect to Figure 4. And we will consult with the editor and adjust.
Point 3: Table 3 overlaps with the line numbers and is difficult to see the details in the table
Response 3: As you pointed out, the ruled lord on the right overlaps the line number and is difficult to see. We are trying to fix, but it is quite difficult to fix. In this regard, we will consult with the editor and adjust.
Point 4:It would be beneficial to perform descriptive statistics for most of the data in the tables
Response 4: We added a commentary to all tables.
Point 5:There are several grammar issues throughout the manuscript
Response 5:We got English calibration form two people who got a PhD degree in the United States.
Again, we appreciate all of your insightful comments.
We worked hard to be responsive to them.
Thank you for taking the time and energy to help us improve this manuscript.

Reviewer 2 Report
Authors revealed that EGFR-TKI re-challenge therapy following first-line EGFR-TKI treatment in elderly patients with advanced NSCLC harboring sensitive EGFR mutations was one of the options among the limited safe and effective treatment options for elderly EGFR-positive lung cancer patients. These findings are very meaningful and valuable. Thus, I am sure that this report is suitable for publication to this journal with no revision.
Author Response
Dear Reviewer 2
Thank you very much for reviewing our manuscript.
We instructed correction of chart and correction of English from another reviewer. We will submit this revised version.
Thank you for taking the time and energy.

This manuscript is a resubmission of an earlier submission. The following is a list of the peer review reports and author responses from that submission.